# Phase Angle and Handgrip Strength as a Predictor of Disease-Related Malnutrition in Admitted Patients: 12-Month Mortality

**DOI:** 10.3390/nu14091851

**Published:** 2022-04-28

**Authors:** Rocío Fernández-Jiménez, Lara Dalla-Rovere, María García-Olivares, José Abuín-Fernández, Francisco José Sánchez-Torralvo, Viyey Kishore Doulatram-Gamgaram, Agustín M. Hernández-Sanchez, José Manuel García-Almeida

**Affiliations:** 1Departmento de Endocrinologia y Nutrición, Quironsalud Málaga Hospital Av. Imperio Argentina, 29004 Málaga, Spain; lara92net@gmail.com (L.D.-R.); mery.garcia.96@gmail.com (M.G.-O.); jose.abuin.fdez@gmail.com (J.A.-F.); jgarciaalmeida@gmail.com (J.M.G.-A.); 2Unidad de Gestion Clínica de Endocrinología y Nutrición, Hospital Universitario Virgen de la Victoria de Málaga, 29010 Málaga, Spain; 3Instituto de Investigación Biomédica de Málaga-IBIMA, 29010 Málaga, Spain; fransancheztorralvo@gmail.com (F.J.S.-T.); viyu90@hotmail.es (V.K.D.-G.); 4Unidad de Gestión Clínica de Endocrinología y Nutrición, Hospital Regional Universitario de Málaga, 29010 Málaga, Spain; 5Departamento de Medicina y Dermatología, Facultad de Medicina, University of Malaga, 29010 Málaga, Spain; 6Departamento de Hematología, Quironsalud Málaga Hospital Av. Imperio Argentina, 29004 Málaga, Spain; agustin.hernandez@quironsalud.es

**Keywords:** phase angle, malnutrition, admitted patient, assessment tools, mortality

## Abstract

Background: Phase Angle (PhA) value measured by bioelectrical impedance analysis (BIA) could be considered a good marker of the patient’s cell mass and cellular damage. Various studies have shown that the value of PhA is associated with an increased nutritional risk in several pathologies. However, not many studies have focused on the use of PhA as a screening tool in admitted patients. The aim of this study is to evaluate the prognostic value of PhA to determine disease-related malnutrition (DRM) and the risk that this entails for mortality and length of stay (LOS). Methods: 570 patients admitted to the hospital for different causes were included in this retrospective observational study. Patients’ nutritional risk was assessed by screening tests such as the Malnutrition Universal Screening tool (MUST) and Subjective Global Assessment (SGA), in addition to non-invasive functional techniques, such as BIA and handgrip strength (HGS), 24–48 h after admission. After performing an SGA as the gold standard to assess malnutrition, PhA and SPhA values were used to determine DRM. Furthermore, both samples: malnutrition status (MS) and non-malnutrition status (NMS) were compared, with SphA-Malnutrition corresponding to a diagnosis of malnutrition. Statistical analysis of the sample was conducted with JAMOVI version 2.2.2. Results: Patients with MS had lower PhA and SPhA than patients with NMS (*p* < 0.001). The ROC curve analysis (AUC = 0.81) showed a cut-off point for MS for PhA = 5.4° (sensitivity 77.51% and specificity 74.07%) and AUC = 0.776 with a cut-off point for SPhA = −0.3 (sensitivity 81.74% and specificity 63.53%). Handgrip strength (HGS) was also observed to be a good predictor in hospitalized patients. Carrying out a comparative analysis between MS and NMS, length of stay (LOS) was 9.0 days in MS vs. 5.0 days in NMS patients (OR 1.07 (1.04–1.09, *p* < 0.001)). A low SPhA-malnutrition value (SPhA < −0.3) was significantly associated with a higher mortality hazards ratio (HR 7.87, 95% CI 2.56–24.24, *p* < 0.001). Conclusion: PhA, SPhA and HGS are shown to be good prognostic markers of DRM, LOS and mortality and could therefore be useful screening tools to complement the nutritional assessment of admitted patients.

## 1. Introduction

Nowadays, malnutrition remains one of the most serious health problems in many groups of patients. It is an important clinical syndrome associated with a high risk of infections, sepsis and mortality [1,2]. Other studies have found a link between malnutrition status and increased length of stay (LOS) in hospital, higher hospitalization costs and impaired recovery [3,4]. Furthermore, there is substantial evidence for the clinical benefits of the early detection of malnourished patients at hospital admission, followed by personalized nutritional interventions [5,6].

Malnutrition is defined, by the European Society for Clinical Nutrition and Metabolism (ESPEN), as “a state resulting from lack of uptake or intake of nutrition causing altered body composition (decreased fat free mass and body cell mass), leading to diminished physical and mental function and impaired outcome from disease” [7]. Although screening techniques are available to detect people at risk of poor nutritional status, there is a need to enhance malnutrition evaluation by recognizing tissue loss, particularly among hospitalized and sick patients [8,9]. While malnutrition is a global problem linked with increased morbidity, mortality, and expenditures, there has been a basic lack of agreement on diagnostic criteria for malnutrition evaluation in clinical settings [10].

The Malnutrition Universal Screening tool (MUST), developed by the British Association for Parenteral and Enteral Nutrition (BAPEN), is a rapid screening tool proven to have content validity (comprehensiveness of the tool), face validity (issues relevant to the purpose of the test) and internal consistency [7,11].

The 2002 guidelines of ASPEN (American Society for Parenteral Nutrition and Enteral) [12] recommend using Desky’s Subjective Global Assessment (SGA) of Desky et al. [13] to establish a nutritional diagnosis. Although this method is accurate, it requires an experienced observer since the nutritional assessment is performed subjectively. 

Bioelectrical impedance analysis (BIA) is a non-invasive tool for estimating body composition which is rapid, safe, and affordable. The phase angle (PhA), measured by BIA, has been interpreted as a measure of membrane integrity and body cell mass and applied as a nutritional diagnostic tool in various patient groups [14,15]. Recent research indicates that low PhA levels are related to malnutrition risk, higher morbidity and mortality in individuals with renal illness, cancer, and surgery [16,17]. Although the association between the PhA and nutritional risk has been well established, the reference or cut-off values for predicting the risk of malnutrition in the hospitalised population are not. Previous research has demonstrated the efficacy of the PhA assessment in predicting survival in a variety of clinical circumstances, as well as nutritional status and illness progression [18]. Low PhA readings have been related to reduced muscular function and poor mortality in critically sick patients, as well as with longer hospitalization [19,20].

However, despite a consistent body of evidence for the clinical value of PhA assessment, few studies have investigated the predictive role of PhA in the outcomes for admitted patients. The primary aim of the present study was therefore to evaluate the prognostic value of PhA assessment to detect disease-related malnutrition (DRM) in hospitalized patients.

## 2. Materials and Methods

### 2.1. Setting Study

In this single-center, retrospective observational study, a sample of patients was assessed within 24–48 h of admission to Quironsalud Malaga Hospital for different reasons. BIA, anthropometric measurements, and assessment tests were performed between January 2019 and June 2021. Pediatric patients, pregnant women and short stay patients were excluded from the study. All patients were evaluated with SGA [13], which was used as a gold standard to diagnose malnutrition.

This study was approved by the Ethics Committee of the Regional Malaga Hospital (2758-N-21). All patients participating in our study met the inclusion criteria (not more than 72 h after admission, agreement to participate in the study by accepted informed consent), and none of the exclusion criteria (participation declined or inability to perform measurement by BIA for reasons related to: ethnicity, extensive skin lesions, extravasation of fluids through the route and local hematomas, amputation etc.) A flow chart diagram shows the patient selection process for our study (Appendix A, Figure 1).

### 2.2. Phase Angle and Other Parameters of Bioelectrical Impedance Vector Analysis (BIVA)

Within the first 24–48 h following hospital admission, patients’ PhA was assessed. Whole-body bioimpedance measurements were taken using a 50 kHz phase-sensitive impedance analyzer (BIA 101 Whole Body Bioimpedance Vector Analyzer (Akern^®^, Pontassieve, Italy)) that infuses 800 A [21,22] through tetrapolar electrodes on the right hand and foot. PhA is represented in degrees as arctan (Xc/R) × (180°/π). An individual standardized PhA value (SPhA) was determined from the sex- and age-matched reference population value by subtracting the reference PhA value from the observed patient PhA value and dividing the result by the respective age- and sex-reference standard deviation (SD) [23].

To minimize measurement variability, we employed a consistent quality assurance process. Daily evaluations of the BIA instrument’s technical correctness were conducted using a precision circuit supplied by the BIA device supplier (Akern). All R and capacitance values were consistently measured within a factor of ±1 Ω the 385 Ohm reference value. Additionally, we examined the repeatability of the BIA measurements in vivo and discovered that the coefficients of variation (CV) for R (resistance) and Xc were from 1 to 2 percent (reactance).

At admission and before BI measurement, body weight and standing height were assessed; weight was obtained using a scale with a sensitivity of 100 g; and height was determined using a laser height rod with a sensitivity of 2 mm. The patient was supine on a hospital bed during all BI measures. Considering that fluid changes occur during the transition from standing to recumbency and have a direct effect on R and Z values, the patient was supine for five minutes before BI measurements in order to stabilize BIA values (±2 Ω for R and ±1 Ω for Xc). BIVA places a premium on the position of the impedance vector, which is derived from R and Xc values normalized to body height (H, m), on the R/Xc graph in relation to tolerance ellipses generated from a specific gender and a healthy reference population (e.g., 50, 75, and 97 percent comparable to 1, 2, and SD) [22]. Patients’ BI measurements were normalized for gender and age using data from healthy Italian adults [21,24].

### 2.3. Anthropometry and Clinical Variables

Demographic characteristics, comorbidities, MUST and SGA test, clinical and anthropometric data and other nutritional measurements were compiled. A trained nutritionist visited the participants in their rooms and formulated a risk of malnutrition score (MUST) based upon current body mass index (BMI), known weight loss and the presence of acute disease/no nutritional intake for 5 days and classified patients as 0 (no-risk), 1 (moderate-risk) and 2 (high-risk). Patients were also classified by means of the SGA into one of three categories: (A) well-nourished; (B) moderately malnourished; or (C) severely malnourished.

Furthermore, we tested handgrip strength (HGS) using a JAMAR hand dynamometer (Model BK-7498, Fred Sammons Inc., Brookfield, IL, USA). Grip strength was measured in a seated position with the elbow flexed at 90 degrees. Patients were instructed to perform three maximal isometric contractions with brief pauses between measurements and the median value was recorded.

### 2.4. Clinical Outcomes

The main endpoint was to assess the prognostic value of SPhA for LOS in hospitalized patients. The secondary end-point was to determine the predictive value of SPhA and HGS for mortality. LOS was calculated by counting the days from hospital admission to discharge, expressed in days. Mortality was defined as death at one year of follow-up during admission or after discharge. 

### 2.5. Sample Size Calculation

We tested the hypothesis that PhA and SPhA used as malnutrition criteria were independent predictors of 12-month mortality in admitted patients. We calculated the sample size using the findings of Garlini et al. [18], with different predictors identified from previous studies. Thus, for an alpha error of 0.05, a power of 80% and a loss rate of 10%, a minimum of 507 patients were needed to attain sufficient power. We therefore aimed to recruit 570 patients. 

### 2.6. Statistical Analysis

Data analysis was mainly carried out using the JAMOVI program (version 2.2.2 macOS). We used descriptive statistics to characterize our patient cohort. Normality of the distribution of quantitative variables was verified by the Shapiro–Wilk test. Descriptive statistics were used for the analysis of categorical variables (absolute and relative frequency) and quantitative variables (mean and SD or median and interquartile range). Clinical data and BIA values between MS and NMS were compared using the Student *t*-test, Mann–Whitney *U* test, or the Chi-squared test.

Evaluations of the diagnostic performance of PhA, SPhA and HGS to detect malnutrition were based on the receiver operating characteristic (ROC) curves and the area under the curve (AUC). We estimated the accuracy of these measurements using AUC by plotting sensitivity versus 1-specifity. ROC curves were used to determine the optimal cut-off values by finding the point of convergence for the greatest sensitivity and specificity. The area under the curve (AUC) indicates the discriminative power of the test. Statistical significance was set at *p* < 0.05.

The Kaplane Meier product-limit estimator at 12 months was used to calculate the cumulative probability of death, to estimate survival and to evaluate the difference among the SPhA-malnutrition and SPhA non-malnutrition cut point. The Kaplane Meier survival curves were compared using the log-rank (Mantel-Cox) test. The time of origin was the admission day and the event was defined death and all cases were censored at their last observation. Differences were considered statistically significant with *p* < 0.05. 

Cox proportional-hazards regression was used to assess the relationship between BIA and mortality in admitted patients. Hazard ratios (HR) and their 95% confidence intervals (CI) were calculated. We used a multivariate model with SPhA-Malnutrition, HGS-Malnutrition, Sex, Age, BMI and Hydration as variables.

Our research compares the values of PhA and SPhA with established indicators of prognosis of admitted patients (SGA and handgrip). 

## 3. Results

A total of 570 patients were admitted to the hospital for different causes. According to hospitalization settings, there were 137 internal medicine patients (24.1%), 81 surgery patients (14.2%), 53 digestive patients (9.3%), 57 cancer patients (10%), 96 hematological patients (16.9%), 37 neurological patient (6.5%), 19 pneumology patients (3.3%), 17 trauma patients (3%), and 72 patients admitted for other causes (12.7%). The mean age was 65.0 years; 266 patients were male (46.7%), while 304 were female (53.3%); 307 patients were at risk of moderate or severe malnutrition (MUST) (53.9%) and the other 263 (46.1%) were not. According to SGA, 329 patients had moderate or severe malnutrition status (57.7%) and 241 patients did not (42.3%). Demographic characteristics, anthropometric measurements, functional test results and patients’ measurements are shown in Table 1.

### 3.1. Associations between PhA and HGS and Malnutrition Screening Tools

Malnutrition was evident in 57.7% of the admitted patients as per the malnutrition assessment (SGA B and C). The multivariate odds ratio of PhA was 0.36 (95% CI = 0.26–0.47, *p* < 0.001) and of HGS was 0.97 (95% CI = 0.94–1.00, *p* = 0.029) as predictors of DRM by SGA criteria. The multivariate odds ratio for PhA and HGS as MUST criteria were 0.38 (95% CI = 0.29–0.50, *p* < 0.001) and 0.98 (95% CI = 0.96–1.01, *p* = 0.0229), respectively (Figure 1).

### 3.2. Optimal Variable Cut-Off Values to Detect Malnutrition in Admitted Patients

Using ROC curves we determined the PhA, SPhA, BCM and HGS value cut-off points for predicting DRM (Figure 2). ROC curve analysis showed that PhA had a significant excellent discriminative ability to detect malnutrition among admitted patients (AUC = 0.835), overall PhA for malnutrition diagnosis was 5.4° (sensitivity 74.69% and specificity 78.42%). On the other hand, the PhA cut-off for DRM diagnosis in male patients was 5.4°, AUC= 0.851 (sensitivity 82.11% and specificity 77.96%). However, female patients had a lower PhA cut-off of 5.3°, AUC = 0.815 (sensitivity 70.34% and specificity 63.53%) (Table 2). Thus, SPhA may be useful as a global cut-off point for all patients screened at admission. For SPhA, AUC was 0.776 and an SPhA value of −0.3 was the most sensitive (81.74%) and specific (63.53%) prognostic factor for malnutrition risk (SPhA-malnutrition).

We also compared these data with the cut-off points for malnutrition prediction using ROC curves for HGS and BCM. For HGS, we found a cut-off point for detecting malnutrition under 27 kg (HGS-malnutrition), (AUC = 0.710; sensitivity 67.61% and specificity 67.48%), but we observed significant differences between males and females. Hence, in males the cut-off point for HGS to detect malnutrition was 34 kg (AUC 0.759; sensitivity 75.64% and specificity 64.94%) while in females it was 19 kg (AUC = 0.709, sensitivity 76.56% and specificity 55.81%). For BCM, the cut-off point was 23.6 kg to detect malnutrition (AUC = 0.810, sensitivity 78.84% and specificity 71.04%) and a significant difference was also observed between males and females. In males, the cut-off point for BCM was 30 kg (AUC 0.857, sensitivity 73.98% and specificity 80.24%), and in females it was 21.7 kg (AUC = 0.832, sensitivity 76.27% and specificity 76.88%). The SPhA-malnutrition showed the highest sensitivity at predicting DRM compared to the other parameters (Figure 2).

### 3.3. Prognostic Factor SPhA-Malnutrition and 12-Month Mortality

The wide distribution of individual impedance point vectors of the admitted patients shows a pattern of vector distribution in quadrants based on their SPhA-Malnutrition and mortality characteristics. Patients with malnutrition status (MS) and non-survival patients (NSP) are grouped in the lower right quadrant, corresponding to high mortality frequencies. This quadrant encompasses patients with decreased cell mass and hyperhydration (Figure 3).

Using the SPhA-Malnutrition cut-off point as a criterion of DRM, we can observe significant differences between MS and NMS patients (Table 3). MS patients’ age is greater (68.0 vs. 61.0 years), and they are predominantly male (53.8%), with a lower weight (67.5 vs. 71.0 kg). MS patients have a lower PhA (4.0° vs. 5.9°), BCM (19.1 vs. 26.4 kg), and HGS (24.0 vs. 28.0 kg). Outcomes show a longer stay in MS patients (9.0 versus 5.0 days) and higher mortality at one year (30.0 versus 3.1%).

We used a 6-component model multivariate analysis to evaluate the utility of bioelectrical parameters as prognostic indicators of mortality in admitted patients (Table 4). We found that a low SPhA-Malnutrition value (SPA < −0.3) was significantly associated with a higher mortality hazards ratio (HR 7.87, 95% CI 2.56–24.24, *p* < 0.001). This trend was also maintained in the models adjusted for confounding variables. Likewise, hydration status and HGS were associated with an increased mortality risk in the crude model (hydration HR: 1.12 (1.04–1.21), *p* = 0.003) and (HGS HR: 3.51 (1.51–8.19), *p* = 0.004)) with this relationship being maintained in the adjusted models. Kaplan–Meier survival curves for SPhA-malnutrition and HSG-malnutrition are illustrated in Figure 4. The log-rank test revealed significant differences between the curves (*p* < 0.001). 

The median of LOS was 7.0 days (NMS 5.0 vs. MS 9.0 days) and a low SPhA-Malnutrition value increased the risk of prolonged hospital stay by around 7% (OR 1.07 (1.04–1.09, *p* < 0.001). The association between LOS and SPhA-Malnutrition was analyzed using multivariate linear regression models. A significant association was found in both the crude and adjusted models for SPhA-malnutrition and LOS, β = 3.081, (95% CI, 1.368–6.2074), *p* = 0.002. We also found an association between LOS and SGA in the adjusted models β = 2.776 (0.975–5.727) *p* = 0.006 (Table 5).

## 4. Discussion

We observed that PhA may be a helpful predictor of DRM, 12-month mortality, and length of stay in admitted patients in the current study. This connection might be explained by malnutrition’s detrimental effect on clinical outcomes [25]. Prognostic evaluation of hospitalized patients is both a problem and an opportunity to enhance care quality, although in-hospital signs remain few. PhA is a direct result of BIA. It represents the link between reactance and resistance by measuring the resistance of cellular membranes to current and the resistance of bodily fluids to the current. This indicator is strongly associated with health and nutritional status because it reflects both the integrity of cellular membranes and the distribution of intracellular water [18,26]. 

BIA is able to indirectly estimate body composition, representing a useful and non-invasive technique for nutritional assessment in the admitted patient. PhA is inversely correlated with SGA score [27], which reflects a higher nutrition risk in individuals with a lower PhA.

Anthropometric measurements that reflect muscle mass, such as arm and calf circumference, also show relations with the complications of the disease [28]. These aspects could explain the role of PhA as a prognostic marker for mortality, considering the association between nutrition risk, low muscle mass, and severity of disease in this outcome.

Our results reinforce the association found in the literature between PhA and HGS parameters using the different malnutrition screening tools. Patients with SGA (B and C) showed an increased risk of malnutrition (OR = 0.36 for PhA and OR = 0.97 for HGS) [18,29].

The cut-off values for PhA to detect malnutrition ranged from 4.73° to 6° in the different studies [30,31,32,33]. The cut-off values for PhA obtained by ROC analysis in our sample were <5.3° in females and <5.4° in males, which are similar to other cohorts that proposed a cut-off point without considering differences between sexes [34]. Although BIA-derived PhA has been used as a nutritional assessment tool in patients, a specific cut-off level is required to help clinical nutritionists to classify admitted patients as either well-nourished or malnourished. 

The choice of the cut-off value is determined by the need to match the sensitivity and specificity of the traditional nutritional tests. The aim of our study was to investigate the link between BIA-derived PhA as an indicator of nutritional status and SGA in admitted patients. We found that it could be recommendable to use different cut-off values for males and females rather than a single cut-off value for all patients. PhA was the most sensitive objective method for diagnosing malnutrition in the group of admitted patients (sensitivity 82.11% in males, 70.34% in females). Our findings are consistent with those reported by another group of researchers who evaluated PhA versus SGA in 73 patients with colorectal cancer [16]. In a study on hospitalized geriatric patients, Dogan Varan et al. [33] observed a PhA cut-off value for malnutrition risk of 4.7° (sensitivity 79.6%, specificity 64.6%) without differences by sex.

ROC analysis provides guidelines to determine the limits of an optimal cut-off level for any diagnostic test according to its clinical context. In our study, we evaluated the optimal cut-off point for phase angle levels used as a nutritional assessment tool in admitted patients. Since malnutrition is one of the main causes of morbidity and mortality in these patients, nutritional diagnosis and treatment could be highly valuable to be able to correctly identify a large percentage of malnourished patients (with a high sensitivity), in spite of this occurring at the expense of reduced specificity (a high false positive rate). In these situations, the most sensitive cut-off point should be chosen. Since PhA depends on sex and age, for a global evaluation of the parameter it is more appropriate to use SPhA, which corrects for these factors. In our study, we found that the cut-off point for SPhA (<−0.3) was more sensitive than the cut-off point for PhA to determine malnutrition (sensitivity 81.74% vs. 74.69%). In another study, Kyle at al. reported PhA cut-offs of 5.0 ° for men with a 70.0% sensitivity and 4.6° for women with 58.1% sensitivity, suggesting malnutrition risk in admitted patients [35]. 

When analyzing BIA measurements in our cohort, lower PhA values were observed in MS with a statistically significant difference between the groups (MS 4.0° (3.4°–4.6°) vs. NMS 5.9° (5.3°–6.7°); *p* ≤ 0.001). The same difference was observed in SPhA (MS −1.8 (−2.8, −1) vs. NMS (+0.9 (0.2–1.8); *p* ≤ 0.001). In a study carried out in cancer patients by Norman et al., ROC analysis revealed that the SPhA with an AUC of 0.734 performed better than SGA (AUC: 0.697) [26]. Moreover, different studies have proposed the value of BIA measurements such as SPhA in assessing nutritional status and predicting survival in several clinical conditions [18,36]. 

Previous studies have also evaluated other nutritional markers such as HGS as predictors of malnutrition. Sostisso et al. showed that the best cut-off point for HGS for men was 23.5 kg (sensitivity 70%; specificity 70%) and for women it was 14.5 kg (sensitivity 70%; specificity 50%) [29]. In our study, the cut-off point for HGS for males was 34 kg (sensitivity 75.6%; specificity 64.94%) and for females it was 19 kg (sensitivity 76.56%; specificity 55.81%). We also found that the cut-off point for BCM differed between the sexes, but previous studies evaluating the association of BCM with malnutrition in 279 admitted patients reported inconsistent results [37]. 

Previous studies evaluating the value of PhA as a prognostic factor in admitted patients reported inconsistent results in relation to its association with clinical outcomes. In a sample of 90 patients with COVID-19, a low PhA was not associated with LOS or with clinical outcomes [25]. Saueressig et al. performed BIA assessment in a sample of 97 patients with decompensated cirrhosis and reported that these had a higher risk of 6-month mortality (HR = 3.44; 95% CI, 1.51–7.84; *p* = 0.003), and each increase of 1° in PhA values was associated with a 53% reduction in 6-month mortality risk [38]. Recently, in a COVID-19 study a cut-off value of 3.95° was obtained as a predictor of 90-day mortality in a sample of 127 patients; 23% required ICU admission during hospitalization [39].

In our sample, an SPhA of −0.3 was found to be the diagnostic criterion of malnutrition to predict mortality (HR 7.78 (2.56–24.24, *p* < 0.01) adjusted by sex, age, BMI and hydration. This differs from the findings presented by Paiva et al., who reported a median SPhA of −1.65, with a higher mortality rate in patients with a low SPhA (RR 3.12 CI: 2.03–4.79; *p* < 0.001) [40]. In the multivariate analysis model, the association between HGS-malnutrition was not significant. 

Another approach is to use the RXc graph as a qualitative and semiquantitative representation of differences in an individual’s cell mass and hydration status [41]. This is fundamental to demonstrate the independent roles of phase angle and vector position in the assessment of nutritional status and hydration in clinical conditions of altered hydration. Phase angle is a composite measure consisting of R and Xc and therefore includes contributions from fluids and cells. It is therefore appropriate to consider hydration per se as a possible confounder of interpretations of phase angle in malnutrition assessment [26]. Patients grouped in the lower right quadrant present high mortality frequencies and a high DRM risk with different hydration situations. 

Furthermore, we also established an association between SPhA-malnutrition and HGS-malnutrition and mortality using a Kaplan–Meier analysis. The worst survival in admitted patients was found in the low SPhA-malnutrition and low HGS-malnutrition; and its prognostic significance was proven in the presence of other clinical indicators of a poor prognostic value such as sex and age. In the graph of the survival curve (Kaplan–Meier), a SPhA < −0.3 and HGS-malnutrition <34 kg (male) and <19 kg (female) grouped the majority of deceased patients, thus constituting significant predictors of mortality. These results are also similar to those reported in other studies in different pathologies [18].

Moreover, it is also important to highlight that the hospital outcome parameter prolonged LOS can be caused by several factors. In fact, longer LOS in malnourished patients has been reported elsewhere. In an Australian study of 819 patients conducted in two teaching hospitals, malnutrition was also associated with an LOS 1.5 times longer than in well-nourished patients [42].

There are several limitations to this study. First, the fact that the sample was obtained from a single hospital, in addition to the retrospective study design, could lead to concerns regarding the difficulties in generalizing the conclusions. Secondly, long-term survival outcomes are not separated by reason of death, because these data were not available. Thirdly, in our study the BIA was performed with AKERM 101 so results could differ from those obtained using other devices. 

## 5. Conclusions

In summary, our study suggests that BIA parameters and SPhA-Malnutrition are potential indicators of nutritional status and mortality risk in general in admitted patients. This biological marker could therefore be incorporated among the screening tools and mortality risk assessment in this population. Patients with low SPhA values (<−0.3) and PhA values (<5.3° in females and <5.4° in males) could therefore receive special nutritional attention. SPhA-Malnutrition is also a predictor of 12-month mortality and is a good predictor of LOS. However, more studies are needed to elucidate the impact of nutritional therapy on BIA parameters and clinical outcomes.

## Figures and Tables

**Figure 1 nutrients-14-01851-f001:**
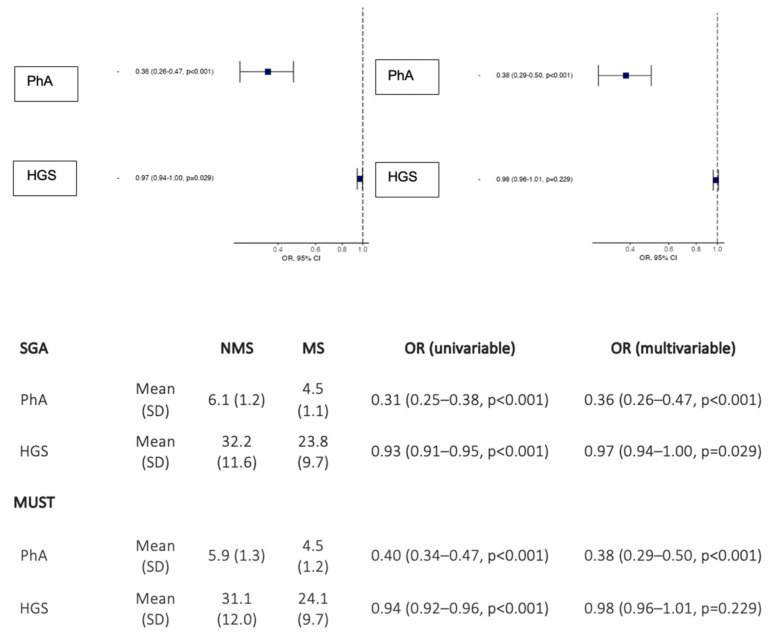
Associations of PhA and HGS with malnutrition tools. Abbreviations: NMS (non–malnutrition status); MS (malnutrition status); OR (odds ratio).

**Figure 2 nutrients-14-01851-f002:**
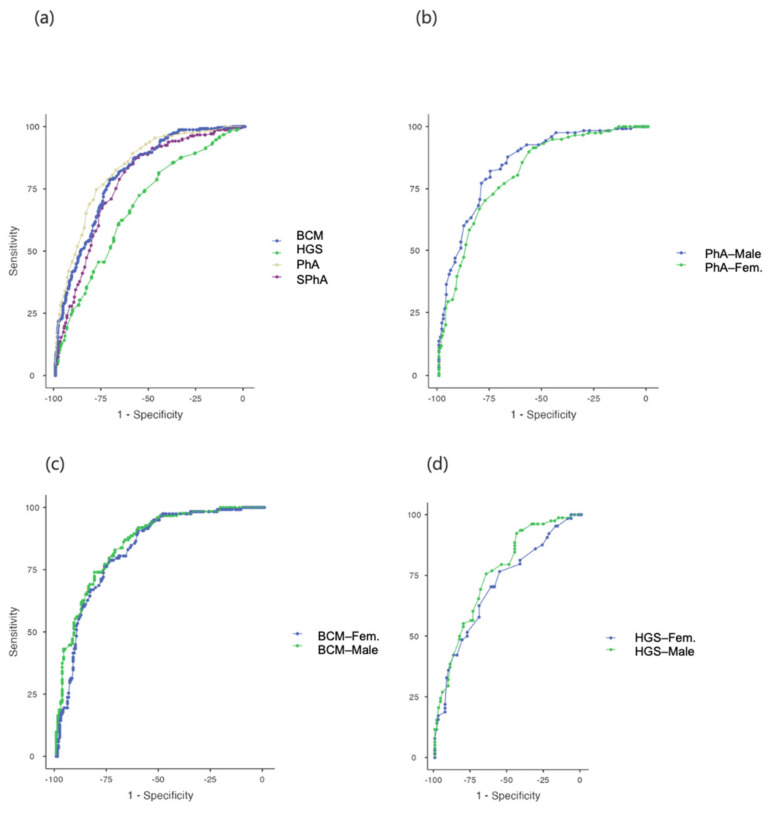
ROC−curve analyses for variables to detect DRM in admitted patients. (**a**) Overall ROC–curve analysis for PhA, BCM, SPhA and HGS; (**b**) ROC−curve analysis for PhA by gender; (**c**) ROC−curve analysis for BCM by gender; (**d**) ROC−curve analysis for HGS by gender to detect DRM.

**Figure 3 nutrients-14-01851-f003:**
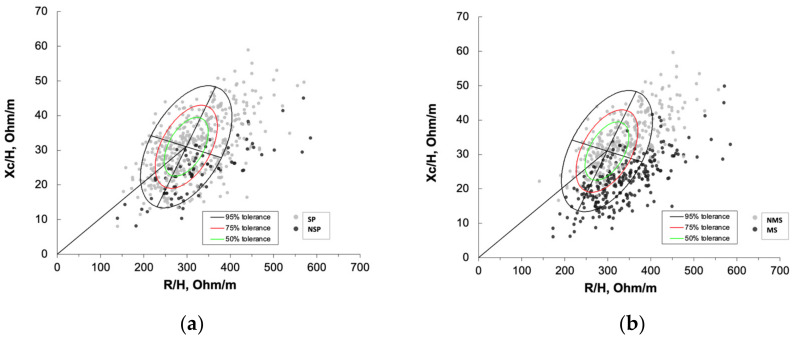
The distribution of impedance point vector of the admitted patients. Representing values of R/H and Xc/H in patients. (**a**) Green point, Survival patients (SP) and black point, non-survival patients (NSP). (**b**) Green point, Malnutrition status (MS) and black point, non-malnutrition status (NMS) using SPhA-malnutrition as a DRM diagnostic parameter. Bioelectrical values of malnutrition: survival patients (n = 484), non-survival (n = 86) and R: resistance (Ohm); Xc: reactance (Ohm); H: height (m); (R/H) and (Xc/H): R/H and Xc/H standardized for sex and age using bioelectrical Italian standards. The bioelectrical impedance vector distribution analysis shows a situation of inflammation and cellular injury associated with malnutrition. The lower right quadrant encompasses patients with decreased cell mass and hyperhydration, most of the deceased patients.

**Figure 4 nutrients-14-01851-f004:**
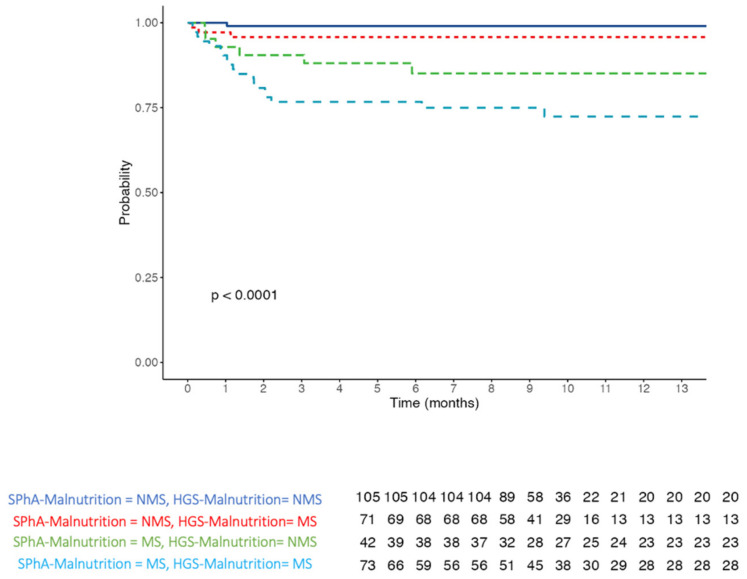
Kaplan–Meier survival curves of patients in groups with low or high SPhA-Malnutrition and HGS-Malnutrition. The table at the bottom indicates the number of surviving patients in each group corresponding to the intervals in the graph. Abbreviations: malnutrition status (MS)—non-malnutrition status (NMS).

**Table 1 nutrients-14-01851-t001:** Demographic parameters, nutritional tools, BIVA, Functional test and Outcomes results by gender.

*Parameters*	*Total*	*Female Patients*	*Male Patients*	*p **
** *Demografics* **				
* ^N^ *	570	304	266	
*Age (years)*	65.0 (53.0–74.0)	64.0 (50.0–75.0)	65 (55.3–73.0)	*p* = 0.498
*Height (cm)*	167.0 (160.0–172.0)	161.0 (156.0–165.0)	172.0 (168.0–178.0)	***p* < 0.001**
*Weight (kg)*	70.0 (60.0–81.1)	63.1 (55.0–74.0)	75.0 (68.0–86.0)	***p* < 0.001**
*Loss Weight (%)*	2.12 (0.0–8.51)	0.9 (0.0–8.1)	3.2 (0.0–8.6)	***p* = 0.015**
*BMI (kg/H^2^)*	24.9 (22.0–28.1)	24.2 (21.4–27.5)	25.6 (23.2–28.5)	***p* = 0.001**
** *Nutritional Tools* **				
*SGA*				*p* = 0.114
*A*	241 (42.3%)	118 (38.8%)	123 (46.2%)	
*B*	204 (35.8%)	116 (38.2%)	88 (33.1%)	
*C*	125 (21.9%)	70 (23%)	55 (20.7%)	
*MUST*				*p* = 0.656
0	263 (46.1%)	136 (44.7%)	127 (48.3%)	
1	123 (21.6%)	70 (23.0%)	51 (42.9%)	
2	184 (32.3%)	98 (32.3%)	86 (46.8%)	
** *BIVA* **				
PhA (°)	5.1 (4.1–6.1)	4.9 (3.98–5.8)	5.45 (4.3–6.57)	***p* < 0.001**
*SPhA*	1.97 (−1.5–1.1)	0.0 (−1.1–1.42)	−0.4 (−1.9–0.77)	***p* < 0.001**
*BCM (kg)*	23.6 (18.7–29.6)	20.4 (16.9–24.4)	29.2 (22.7–35.1)	***p* < 0.001**
** *Functional test* **				
*HGS (kg)*	26.0 (19.0–35.0)	20.0 (17.0–25.8)	35.0 (27.0–40.0)	***p* < 0.001**
** *Outcomes* **				
*Long stay (days)*	7.0 (3.0–12.3)	7.0 (7.0–12.0)	7.0 (7.0–14.0)	*p* = 0.871
*Death, n (%)*	86 (15.1%)	42.0 (13.8%)	44.0 (16.5%)	*p* = 0.450

* *p* for comparison by gender patients. BMI body mass index; H: height; MUST (Malnutrition Universal Screening tool). Subjective Global Assessment (SGA); PhA: phase angle; SPhA: standardized phase angle; BCM: body cell mass, HGS: handgrip strength.

**Table 2 nutrients-14-01851-t002:** Diagnostic accuracy of different parameters to detect malnutrition in admitted patients.

Parameters	AUC	Cut-Off Point	Sensitivity	Specificity	PPV	NPV
**PhA (°)**	0.835	5.4°	74.69%	78.42%	71.71%	80.88%
**Male**	0.851	5.4°	82.11%;	77.96%	74.26%	83.08%
**Female**	0.815	5.3°	70.34%	63.53%	66.94%	80.56%
**SPhA**	0.776	−0.3	81.74%	63.53%	62.15%	82.61%
**HGS (kg)**	0.710	27	67.61%	67.48%	64.43%	70.51%
**Male**	0.759	34	75.64%	64.94%	68.6%	72.46%
**Female**	0.709	19	76.56%	55.81%	56.31%	76.19%
**BCM (kg)**	0.810	23.6	78.84%	71.04%	66.67%	82.04%
**Male**	0.857	30.0	73.98%	80.28%	76.47%	78.08%
**Female**	0.832	21.7	76.27%	76.88%	67.67%	83.63%

**Abbreviations:** AUC (area under the curve); PPV: Positive predictive value. NPV; negative predictive value; PhA: phase angle; SPhA: standardized phase angle; BCM: body cell mass, HGS: handgrip strength.

**Table 3 nutrients-14-01851-t003:** Demographic parameters, nutritional tools, BIVA, Functional test and Outcomes results by SPhA-Malnutrition.

Parameters	NMS	MS	*p* *
**Demografics**			
N	317	253	
Sex n (%) Female	187.0 (59.0%)	117.0 (46.2%)	*p* ≤ 0.001
Male	130.0 (41.0%)	136.0 (53.8%)	*p* ≤ 0.001
Age (years)	61.0 (49.0–70.3)	68.0 (60.0–77.0)	*p* ≤ 0.001
Height (cm)	165 (160.0–172.0)	168 (160.0–174.0)	*p* = 0.644
Weight (kg)	71.0 (60.5–83.0)	67.5 (59.0–79.5)	*p* = 0.018
Loss Weight (%)	0.09 (0.0–5.8)	5.53 (0.0–13.2)	*p* ≤ 0.001
BMI (kg/H^2^)	25.3 (22.7–28.7)	24.4 (21.4–27.3)	*p* = 0.002
**Nutritional Tools**			
VSG			*p* ≤ 0.001
A	197(62.1%)	60 (23.7%)	
B	90 (28.4%)	64 (25.3%)	
C	30 (9.5%)	129 (51.0%)	
MUST			*p* ≤ 0.001
0	203 (64%)	60 (23.7%)	
1	59 (18.6%)	64 (25.3%)	
2	55 (17.4%)	129 (51%)	
**BIVA**			
PhA (°)	5.9 (5.3–6.7)	4.0 (3.4–4.6)	*p* < 0.001
SPhA	0.9 (0.2–1.8)	−1.8 (−2.8–−1)	*p* < 0.001
BCM (kg)	26.4 (22.5–32.9)	19.1 (15.6–24)	*p* < 0.001
**Functional test**			
HGS (kg)	28.0 (19.0–38.3)	24.0 (18.0–33.0)	*p* = 0.028
**Outcomes**			
Long stay (days)	5.0 (3.0–9.0)	9.0 (5.5–17.0)	*p* < 0.001
Death, n (%)	10.0 (3.1%)	76.0 (30%)	*p* < 0.001

* *p* for comparison by gender patients. NMS: Non malnutrition state; MS: malnutrition state; BMI body mass index; H: height; PhA: phase angle; SPA: standardized phase angle; BCM: body cell mass; HGS: handgrip strength.

**Table 4 nutrients-14-01851-t004:** Model multivariate analysis to evaluate the utility of the bioelectrical parameters as prognostic indicators of mortality in admitted patients.

Dependent: Surv (Mytime, Myoutcome)		All	HR (Univariable)	HR (Multivariable)
SPhA-Malnutrition	MS	163 (100.0)	-	-
	NMS	107 (100.0)	9.62 (3.35–27.66, *p* < 0.001)	7.87 (2.56–24.24, *p* < 0.001)
HGS-Malnutrition	NS	137 (100.0)	-	-
	MS	133 (100.0)	3.51 (1.51–8.19, *p* = 0.004)	2.23 (0.92–5.41, *p* = 0.076)
Sex	Female	134 (100.0)	-	-
	Male	136 (100.0)	0.94 (0.46–1.93, *p* = 0.874)	0.85 (0.40–1.80, *p* = 0.665)
Age	Mean (SD)	61.9 (14.9)	1.03 (1.00–1.05, *p* = 0.069)	1.01 (0.98–1.04, *p* = 0.416)
BMI	Mean (SD)	25.5 (4.9)	0.94 (0.87–1.02, *p* = 0.157)	0.98 (0.91–1.07, *p* = 0.705)
Hydration	Mean (SD)	74.4 (5.7)	1.12 (1.04–1.21, *p* = 0.003)	1.00 (0.92–1.08, *p* = 0.994)

**Abbreviations:** HR (hazard ratio); PhA: phase angle; SPA: standardized phase angle; BMI body mass index.

**Table 5 nutrients-14-01851-t005:** Association between SPhA-malnutrition and LOS.

			95% Confidence Interval			
**Model 1**	**Estimate**	**SE**	**Lower**	**Upper**	**ß**	** *p* **
Intercept ^a^ SPA_Malnutrition: MS-NMS	7.6	0.539	6.54	8.66	14.1	<0.001
	5.32	0.813	3.72	6.92	6.54	<0.001
**Model 2**						
Intercept ^a^ SPhA_Malnutrition	7.139	2.3344	2.543	11.7347	3.058	0.002
MS-NMS	3.7875	1.2292	1.368	6.2074	3.081	0.002
Age	−0.0291	0.0362	−0.1	0.0421	−0.805	0.421
Sex						
Female–Male	0.307	1.0729	−1.805	2.4192	0.286	0.775
SGA:						
B/C–A	3.3511	1.207	0.975	5.7273	2.776	0.006
HGS_Malnutrition						
1–0	0.5995	1.1347	−1.634	2.8335	0.528	0.598
Death						
1–0	1.1102	1.7216	−2.279	4.4995	0.645	0.52

^a^ Represents reference level. All values are expressed as ß (standard error). Model 1: unadjusted. Model 2: adjusted for age, gender, SGA, HGS-Malnutrition and death. *p* < 0.05 SPhA: standardized phase angle; LOS: length of stay; PhA: phase angle; SGA: Subjective Global Assessment; HGS: handgrip strength.

## Data Availability

The data that support the findings of this study are available from the corresponding author, upon reasonable request.

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
