# Peer review of "Phase Angle and Handgrip Strength as a Predictor of Disease-Related Malnutrition in Admitted Patients: 12-Month Mortality"

_nutrients, 2022, doi:10.3390/nu14091851_

Round 1

Reviewer 1 Report

The paper is well motivated and well written.

There are some points which must be address before publication.

1-The kaplane meier product limit estimator is used to measure fraction of patient after treatment but you are assessing it on the admitted patient 24-48 hours. How do you estimate share your data with R codes.

2-A sample is taken and assessed within 24-48 hours is it count in long stay if no/yes, then how can you judge follow-up patient. If you have data of follow up patient also share it kindly.

3-By using malnutrition screening tool (MUST) and subjective global assessment (SGA) all factors is used in your study?? If yes/no, then which criteria is use for including excluding dependent/independent variables and why?

Reviewer 2 Report

As you stated, the identification of malnutrition in the hospital setting is an important issue and there continues to be a dearth of effective malnutrition  screening tools. This is an excellent paper, well-written and of clinical and scientific importance.

I have minor comments only:

Table 1:  What is "VSG"? Should this be "SGA" since SGA doesn't appear in the table but is included in the footnote? Also, recheck the results described in lines 181-184. Those numbers and %'s don't exactly match the results listed in Table 1.

Line 204, the number is stated incorrectly

Appendix A:  Wording can be improved for some of the descriptions; confirm the n listed in the bottom right box is correct or explain it differently (i.e. 631-115 doesn't = 570)
